

# T&C-CROP: Representing mechanistic crop growth with a terrestrial biosphere model (T&C, v1.5): Model formulation and validation.

Jordi Buckley Paules[1], Simone Fatichi[2], Bonnie Warring[3], Athanasios Paschalis[1]

[1]Department of Civil and Environmental Engineering, Imperial College London.
5   [2] Department of Civil and Environmental Engineering, National University of Singapore.
[3] Grantham Institute on Climate Change and the Environment, Imperial College London.

*Correspondence to*: Jordi Buckley Paules (j.buckley21@ic.ac.uk)



## Abstract:

Cropland cultivation is fundamental to food security and plays a crucial role in the global water, energy, and carbon cycles. However, our understanding of how climate change will impact cropland functions is still limited. This knowledge gap is partly due to the simplifications made in Terrestrial Biosphere Models (TBMs), which often overlook essential agricultural management practices such as irrigation and fertilizer application, and simplify critical physiological crop processes.

Here we demonstrate how with minor, parsimonious enhancements to the TBM T&C it is possible to accurately represent a complex cropland system. Our modified model, T&C-CROP, incorporates realistic agricultural management practices, including complex crop rotations, irrigation and fertilization regimes, along with their effects on soil biogeochemical cycling. We successfully validate T&C-CROP across four distinct agricultural sites, encompassing diverse cropping systems such as multi-crop rotations, monoculture, and managed grassland.

A comprehensive validation of T&C-CROP was conducted, encompassing water, energy, and carbon fluxes, Leaf Area Index (LAI), and organ-specific yields. Our model effectively captured the heterogeneity in daily land surface energy balances across crop sites, achieving coefficients of determination of 0.77, 0.48, and 0.87 for observed versus simulated net radiation (Rn), sensible heat flux (H), and latent heat flux (LE), respectively. Seasonal, crop-specific gross primary production (GPP) was simulated with an average absolute bias of less than 10%. Peak season LAI was accurately represented, with an $r^2$ of 0.67. Harvested yields (above-ground biomass, grain, and straw) were generally simulated within 10-20% accuracy of observed values, although inter-annual variations in crop-specific growth were difficult to capture.



## 1. Introduction

## 1.1 Climate Change, food security and the need for process-based crop models.

Understanding the impact of weather and field management on cropland productivity is critical,
not least in the face of mounting challenges such as anthropogenic climate change and shifting
socio-demographics (Godfray et al. 2010; Foley et al. 2011; FAO, 2022; Cammarano et al. 2022;
Wang et al. 2022).  The effects of climate change on both local and global agri-food systems
are expected to increase, with shifts in the frequency, intensity, and timing of droughts and
heatwaves, all posing real threats to crop growth (Ortiz-Bobea et al. 2021;Dury et al. 2022; FAO,
2022; Kim and Mendelshorn, 2023). The effects of climate change on agriculture are set to vary
spatially, with a large degree of heterogeneity between regions (Semenov, 2009; Waha et al.
2013; Ukkola et al. 2020;Moustakis et al. 2021; Slater et al. 2022). Therefore mitigation efforts
will demand a nuanced understanding of processes, causes and ultimately effects. For example,
as a function of anthropogenic emissions, global $CO_2$ is rising roughly uniformly, however its
effect on crop growth dynamics, termed the $CO_2$ fertilization effect, is likely to vary regionally
(McGrath and Lobell, 2013); likely due to complex non-linear interactions between $CO_2$,
temperature, water and nutrient availability. Processes such as the above make the study of
climate-crop interactions particularly interesting, and complex (Lawlor and Mitchel, 1991; Polley,
2002; Fatichi et al. 2016; Cernusak, 2020; Hussain et al. 2021).


One way to address the challenges climate change poses to crops is to deepen our
understanding of climate-crop interactions and their interface with field management practices
through the development of process-based models. A particular strength of this approach is its
potential to enhance our understanding and forecasting capabilities beyond current or past
observations (Boote et al., 2013; Muller and Martre, 2019). Such research is vital to align
agronomic strategies with societal food demands, all whilst promoting environmental
sustainability, as emphasized by Cassman and Grassini (2020).



## 1.2 Diversity in crop models, strengths and limitations

A vast array of models have been developed to capture the interactions between soil, crop, climate and field management practices. It is possible to lump these models into one of three categories; statistical, conceptual or physics based. **Statistical** models are entirely data driven and contain little to no pre-conceived representation of physical processes, they rely on historical data to establish statistical relationships between crop yield and climate variables (e.g. Lobell and Burke, 2010; Gaupp et al. 2019; Van Klompenburg et al. 2020; Ansarifar et al. 2021; Slater et al. 2022). **Conceptual** models represent key physical processes in a simplified fashion which can then be parameterised or calibrated to best fit observational data, an example is Aquacrop (Steduto et al. 2009) but many others crop models have been developed (Di Paola et al. 2016). **Physics based** models codify state of the art understanding of physical laws, such as conservation of energy, water, carbon and momentum, into a crop modelling framework. Examples here include CLM-CROP (Drewniak et al. 2013; Bilionis et al. 2014; Sheng et al. 2018; Boas et al. 2021), JULES-CROP (Osborne et al. 2014; Williams et al. 2017), Gecros (Ingwersen et al. 2018) or ORCHIDEE-CROP (Wu et al. 2016). These physics-based models are built on the latest scientific understanding of soil-plant-atmosphere interactions. They start by resolving photosynthesis and plant energy budgets and incorporate key processes such as water and nutrient uptake, crop phenology, and carbon allocation schemes (Fatichi et al. 2019; Wiltshire et al. 2021; He et al. 2021). A comprehensive review on the respective limitations of different modelling frameworks is provided by Roberts et al. (2019). Comparative studies have shown that, in terms of yield prediction, process-based models are currently less effective than their statistical counterparts (Leng and Hall, 2020). This may be attributed to the higher complexity of physics-based models, where yield is the by-product of multiple processes, and to current data limitations that hinder the proper parameterization and calibration of these models (He et al., 2017).



The question thus arises as to why prioritise further development of physics-based models in agricultural research? Firstly, physics-based models address several limitations inherent to statistical crop models. These limitations include issues such as multicollinearity between climate variables and yield, as well as lack of potential generalizability beyond their calibration envelope. This latter point is crucial, as statistical models rely on historical climate-yield relationships which may not hold true under future climates (Sheehy et al. 2006; Boote et al 2013; Lobell and Asseng, 2017). Secondly, physics based models offer explicit representation of coupled dynamics, including water, carbon and nutrient cycles. These dynamics are expected to be significantly impacted by climate change, making their understanding crucial for accurate crop yield projections and sustainable agricultural management. Lastly, whilst physics-based models do currently face challenges due to data requirements, such as climate forcing and crop-specific traits, this obstacle is expected to diminish over time. The integration of evolving plant databases, such as the TRY database (Kattge et al. 2020), and advancements in remote sensing technologies (Khanal et al, 2020; Wu et al. 2023) are anticipated to yield more comprehensive datasets. This increasing availability of data is likely to enhance the effectiveness and reliability of future physics-based crop models.

## 1.3 Space for a new TBM Crop model, needed developments.

In a bid to better capture the intricacies of cropland dynamics, various previous studies have further developed existing TBMs akin to T&C (Fatichi et al. 2012;2019). Examples include, JULES-CROP (Osborne et al. 2014), CLM-Crop (Drewniak et al. 2013; Bilionis et al. 2014; Sheng et al. 2018; Boas et al. 2021), ORCHIDEE-Crop (Wu et al., 2016) and CARAIB DGVM (Jacquemin et al. 2021). Commonly, model developments in the context of TBMs centre on the introduction of new crop-specific modules, which incorporate crop-specific carbon pools and dynamics alongside harvest indexes and management options. While these past endeavours represent a significant step forward, they often introduce multiple modifications that may not generalize well.





Despite these advancements, there remains a need to improve the integration of crop management practices such as sowing, harvesting, irrigation, and fertilizer application within TBMs. This would more comprehensively capture the coupled dynamics of plant growth and soil biogeochemical cycles, as influenced by crop nutrient uptake and the timing and quantity of NPK fertilizer application. For example, previous work with JULES-CROP (2014) omitted nutrient limitations, while ORCHIDEE-Crop (Wu et al. 2016) addressed nutrient limitation via a simple empirical 0-1 index limiting crop growth. Furthermore, irrigation practices need better incorporation; ORCHIDEE-Crop (Wu et al. 2016) omitted irrigation, while JULES-CROP (Williams et al. 2017) assumed perfect irrigation, neglecting soil moisture as a crop growth stress factor. Additionally, there is a need to transition from empirical harvest indices or harvest-specific carbon pools to a fully integrated mechanistic approach, whereby crop yield is derived from generalizable carbon organ-specific pools being harvested.

Most importantly, the goal of introducing crops into Terrestrial Biosphere Models (TBMs) should be to do so with minimal changes to the existing model structure for natural vegetation, as most physical and biophysical processes are similar. We argue that this can be accomplished without adding additional carbon pools or extensive model modifications and parameter additions. The aim is to demonstrate that accurate crop representation within a TBM can be achieved in a parsimonious manner, avoiding the need for crop-specific parameterizations that are difficult to generalize. This approach differentiates our model from previous formulations.

Our study introduces T&C-CROP to address the aforementioned challenges, building on the success of previous Terrestrial Biosphere Models (TBMs). Previous developments to T&C (Fatichi et al. 2012; 2019) have ensured that an effective representation of crops, irrigation, and fertilizer application can be seamlessly integrated into the established vegetation carbon pool dynamics. This integration links agricultural practices with water and energy budgets, plant growth development, and soil biogeochemical cycling. All enhancements to the original T&C



model involve minimal structural changes. Specifically, only three parameters are added to the
original model, along with irrigation, fertilizer, and the sowing and harvesting dates.

To assess the effectiveness of T&C-CROP, we evaluated model performance in terms of
energy, water and carbon fluxes with on-site eddy covariance data and benchmarked it against
other TBMs with dedicated crop-specific modules at the same sites. We assessed T&C-CROP's
skill in predicting crop yields, specifically examining carbon allocation to various pools, making
good use of detailed harvest data available across the selected sites. The evaluation covers
four fields which employ varied management strategies and operate in diverse climates.

## 2. Materials and Methods

## 2.1 Overview of T&C

T&C is a state-of-the-art terrestrial biosphere model (Fatichi et al. 2012;2019) which resolves
the land surface energy balance, water balance and soil C/N/P/K dynamics. T&C has been
successfully used in several ecosystems globally covering a wide range of scenarios, for
example assessing the impacts of fertilization on grassland productivity in the European Alps
(Botter et al. 2021) or assessing ecohydrological changes after tropical conversion to oil palm
(Manoli et al. 2018). T&C operates across various time scales, tailoring its resolution to the
specific process being resolved. Specifically, the energy budget is resolved at hourly scales,
water and photosynthesis are computed at the hourly scale, with the exception of soil water flow
that uses an adaptive sub hourly step, vegetation carbon pools and soil C/N/P/K dynamics are
resolved at the daily scale. Inputs consist of hourly meteorological data (precipitation,
temperature, wind speed, atmospheric pressure, relative humidity, shortwave and longwave
radiation, atmospheric $CO_2$ concentration). Site parameterization requires site-specific
information including soil texture, and plant specific traits for tailoring the dynamic vegetation
component. T&C does not use predefined plant functional types, but uses a vegetation specific
approach where the model user defines the vegetation/crop in question. T&C can be run as a



plot-scale version, i.e., without an explicit treatment of the topography and lateral fluxes (e.g., Paschalis et al. 2017; Manoli et al. 2018 and this study) or alternatively in a spatially explicit manner, which accounts for complex topography by considering local and remote solar radiation shading effects and lateral transfer of water in the surface and subsurface (e.g., Paschalis et al. 2017; Mastrotheodoros et al. 2020; Paschalis et al. 2022).

The hydrological module of T&C is physics-based and models interception, throughfall, canopy water storage, runoff and soil water dynamics, as well as snow and ice hydrology. Soil water dynamics are represented in the point scale simulations via the 1-D Richards equation. In this study soil hydraulic conductivity alongside the shape of the water retention curve are estimated based on user-defined soil texture; following the Saxton and Rawls pedotransfer function

(Saxton and Rawls, 2006; Paschalis et al. 2022). However, T&C can also use custom water retention curves including the van Genuchten model and more complex soil hydraulic function accounting for soil structural effects (Fatichi et al. 2020). Plant-water uptake is simulated using a sink term, with plant transpiration uptake being proportional to root biomass which decays exponentially with soil depth. Both saturation and infiltration excess mechanisms are used for

runoff generation (Fatichi et al. 2012).

The surface energy balance is resolved by balancing net radiation with latent, sensible and ground heat fluxes. In T&C, we use the two-stream approximation for estimating net shortwave radiation with a canopy being split into a sun and shaded fraction (de Pury and Farquhar, 1997; Wang and Leuning, 1998; Dai et al. 2004). Latent and sensible heat fluxes are parameterized

using the resistance analogue, with aerodynamic, leaf-boundary layer, stomatal, and under canopy air resistances as well as soil resistance all included (e.g. Leuning, 1995; Niyogi and Raman, 1997; Haghigi et al. 2013; Paschalis et al. 2017).

Plant carbon dynamics in T&C are inspired by Friedlingsein et al. (1998) and Krinner et al. (2005). Vegetation is conceptualized using 7 carbon pools for woody vegetation (leaves, living

sapwood, heartwood, dead leaves, roots, carbohydrate reserves and fruits and flowers) and 5



pools for herbaceous species with the sapwood and hardwood carbon pools supressed. Carbon allocation is governed by phenology, environmental stresses, and stoichiometric constraints for C:N, C:P, C:K ratios across all tissues which in turn depend on the potential of plants to acquire necessary macronutrients (NPK) from the ground via root uptake and mycorrhiza symbiosis. In 210 T&C, for extratropical climates we have four phenological stages (dormant, maximum and normal growth, and senescence) defined by temperature, light, water stress and leaf age. Initially, carbon is assimilated via photosynthesis which is based on Farquhar et al. (1980) for C3, and Collatz et al. (1991, 1992) for C4 plants with subsequent adjustments (Bonan et al. 2011) and then scales from leaf to canopy scale according to a two big leaf approach  (Wang 215 and Leuning, 1998; Dai et al., 2004). This approach has the benefit of taking into account the vertical distribution of nitrogen and therefore also of photosynthetic capacity. The CAM photosynthetic pathway is currently not considered. Stomatal conductance follows Leuning (1990; 1995) and has been recently adapted to consider plant hydraulics (Paschalis et al. 2024) although this scheme is not considered here. Any assimilated carbon which is not respired via 220 maintenance and growth respiration, is subsequently partitioned into one of five carbon pools (foliage, living sapwood, roots, carbohydrate reserves or fruits and flowers) via an empirical allocation scheme; largely based on phenological stages and light and water availability. The translocation of carbon between pools is also considered, enabling the depletion of carbon stored as reserves. This better represents the responses of vegetation to stress and changes 225 in phenological stages. Details of plant phenology dynamics are outlined in the supplementary of Fatichi et al. 2012.

The latest version of T&C includes soil carbon and nutrient (nitrogen, phosphorus, and potassium) dynamics (Fatichi et al. 2019). Options for anthropogenic nutrient application (fertilizer), in both mineral and organic forms have been added (Botter et al. 2021). Leaching of 230 dissolved nutrients is also computed by coupling soil biogeochemistry with T&C's soil hydrology module. Specifically, the biogeochemistry module separates plant litter into different pools based on decomposability recalcitrance and account for different soil organic carbon functional pools, as mineral associated, particulate and dissolved organic carbon. Its



decomposition/mineralization depends on the activities of microbial biomass separated between
bacteria and fungi and macraufauna in the soil. NPK cycles (including fertilizer application) are
linked to microbial dynamics and naturally, plant growth. A comprehensive outline of T&C soil
biogeochemistry is provided by Fatichi et al. (2019) and Botter et al. (2021).

## 2.2 From T&C to T&C-CROP

T&C-CROP adds parameterizations designed to enhance the representation of crops within the
T&C model, improving its ability to simulate crop vegetation dynamics. A total of three new
parameters—$SL_{emercrop}$ , a value of additional specific leaf area at leaf emergence; so$_{crop}$, a
parameter to shift carbon allocation; and $Max_{height}$ , maximum crop height—were added to the
original code. Additionally, irrigation, fertilizer (N/P/K), and sowing and harvest dates for each
crop were also included.

Crops, like many plants, exhibit changes in their Specific Leaf Area (SLA) over time (Amanullah,
2015; Li et al. 2023), defined as the leaf area divided by its dry weight (m² kg⁻¹). Early in their
growth stages, leaves tend to have a higher SLA, indicating thinner and cheaper leaves that
facilitate rapid expansion of the leaf canopy and higher photosynthetic rates, essential for early
plant growth post-germination. However, as leaves age, they typically become thicker, resulting
in a lower SLA. To better capture this phenomenon and align with observed trends, we've
implemented a dynamic SLA in T&C-CROP. This dynamic SLA is modelled with a linearly
decaying rate from an initial maximum SLA until the leaf age reaches the value of the
phenological stage of maximum growth, beyond which SLA retains a constant value.

$$SLA_{new} = \begin{cases} SLA + \left(1 - \frac{Age_L}{dmg}\right) * SL_{emercrop}, if\ Age_L < dmg \\ SLA,\ if\ Age_L \geq dmg \end{cases} \qquad [1]$$



Where, *SLA* represents the full grown crop static specific leaf area (m² gC⁻¹), $Age_L$ [days] denotes the age of the leaf in days, $SL_{emercrop}$ is a new parameter representing the additional *SLA* at emergence which can be crop-dependent, and *dmg* signifies the days of maximum leaf growth phenology stage, which is a model parameter. Variable names are intentionally kept identical to model parameters in T&C which can be accesses from our repository (see data availability).

We also aimed to enhance the portrayal of the initial leaf flushing period. At the onset of crop growth, carbon allocation to fruits and flowers is impeded, with newly assimilated carbon instead directed towards leaf development. As the initial leaf flush concludes, carbon allocation shifts predominantly towards the fruits and flower pool with a reference value allocation fraction $f_{fr}$ [-] to this pool, which is significantly higher than for natural vegetation, while allocation to living sapwood is reduced or nullified if the crop does not have a stem component by using a crop specific parameter $so_{crop}$ [-] which is the carbon allocation fraction to stem. These values can be user-defined and crop-specific, but generally for crops $f_{fr}$ is in the order of 0.2-0.5 and $so_{crop}$ in the order of 0.0-0.1.

Typically, photosynthetic efficiency decreases as leaves age. For example, this is the case with wheat (Suzuki et al. 1987). To replicate the rapid drop in late season photosynthesis of senesced leaves, once a leaf's age exceeds a critical threshold (*age_cr* ), the photosynthetic efficiency is reduced as a power law (power of minus eight) of the relative age ($r_{age}$). Where $r_{age}$ is the relative time from leaf onset exceeding *age_cr.*

Additionally, we updated the leaf turnover function, which represents the rate of leaf mortality due to aging. Our update is illustrated below in Eq. 2, where *dla* is the leaf death rate [days⁻¹] due to age, *age_cr* is the critical leaf age (a crop-specific parameter), and *AgeL* [day] is the current average age of the leaf (a prognostic variable). Previously, T&C applied a linear relation for grass and extratropical evergreen trees and a power law for deciduous tree leaves (Fatichi




et al., 2012; 2019). Our modification, in the form of a sigmoidal function (Supplementary 1), ensures that the majority of leaf turnover occurs as leaf age approaches the critical age, and suppresses completely leaf mortality in the early phases, which is more realistic for crops.

$$dla = \left(\frac{1}{age_{cr}}\right) \times \left(\frac{1}{2}\tanh\left(10 \times \left(\frac{AgeL}{age_{cr}}\right) - 7\right) + 0.5\right) \qquad [2]$$

To enable crop representation in T&C-CROP, we have introduced the option of user defined sowing and harvesting dates. In the model, sowing is conceptualized by introducing an initial carbon stock for fine root biomass and non-structural carbohydrates, comparable to typical seed
applications, from which the crops evolve post-germination. Root depth can be parameterized as a function of fine root biomass or fine root growth, if allometric relationships are available, or kept constant if such knowledge is unavailable. After crop establishment, leaf age or environmental stress can trigger crop senescence before harvesting. Additionally, to accommodate multiple crop management practices, users can define the fraction of the crop left
in the field post-harvest. This feature can be tailored to specific crops or management practices, such as leaving stems behind while harvesting only grains. This flexibility allows for a more nuanced representation of different cropping systems and practices within the model.

In summary, while the model structure was modified to better tailor specific leaf area, carbon allocation, leaf turnover, and photosynthetic efficiency of senesced leaves to crop conditions,
we added only two additional parameters—$so_{Crop}$, and $SL_{\_emercrop}$—for these purposes. This approach is much more parsimonious than many other crop implementations in process-based models (e.g., Ingwersen et al. 2018). Additionally, sowing and harvesting dates, the amount of carbon in seeds during sowing (typically 5-20 gC m²), and the maximum crop height, which is the third new parameter, also need to be specified in T&C-CROP.



## 2.3 Simulation Setup

T&C-CROP was run at a plot scale (i.e., neglecting topographic features) and used site-specific hourly meteorological data, retimed from the half-hourly data available from local weather observations (Table 2). In T&C-CROP the partitioning of shortwave radiation to direct/diffuse and different wavelengths such as Photosynthetic Active Radiation (PAR) was done using REST2, as implemented in AWEGEN (Fatichi et al. 2011; Peleg et al. 2017). Site-specific data such as dates of planting/sowing/irrigation/fertilizer application and soil type were obtained either from available literature (references in Table 2) or directly from the site's PI. To balance the soil carbon and nitrogen pools an appropriate spin-up was run, the length required to reach a dynamic steady state was site dependent but normally in the order of 200 years.

T&C-CROP, like T&C does not use generic plant functional types, meaning the user must input plant or crop-specific parameters, some of which are illustrated in Table 1. These were obtained from literature and the TRY database (Kattge et al. 2020; Fraser, 2020). However, the final values used in the model runs were adjusted within a $\pm30\%$ range from the reported values as part of a manual trial and error calibration, necessary to best fit the cultivar type being sown on each site. Temperature and daylength thresholds for phenological changes were retrieved with expert knowledge and manual calibration at each site matching leaf area observations. Furthermore, in T&C-CROP the user inputs the date of sowing, therefore the start date for crop growth is largely prescribed through crop management. Other models such as AquaCrop (Steduto et al. 2009) calculate the sowing date dynamically based on local environmental conditions. This is also possible in T&C-CROP, but for this study, as sowing dates were available at all sites (Supplementary 2), for best realism they were prescribed. Following emergence, plant growth is purely dependent on local climate and environmental conditions. Inputs regarding fertilizer/irrigation application are inputted based on the management log shared by the PI (e.g. Supplementary 3) or where not available we used typical values for the region and crop type.





**CROP MODEL VARIABLES**

| PARAMETER | UNIT | DESCRIPTION |
|---|---|---|
| SL | $m^2$ / gC | Specific leaf area |
| AGE_CR | day | Critical Leaf Age |
| TLO | Celsius | Temperature for leaf onset |
| DMG | day | Days of Max Growth |
| TRR | gC / $m^2$ d | Translocation rate |
| LDAY_MIN | - | Minimum Day duration for leaf onset |
| LTR | - | Leaf to Root ratio maximum |
| VCMAX | µmol $CO_2$/ $m^2$ s] | Maximum Rubisco capacity at 25°C leaf level |
| BFAC | - | Leaf onset water stress threshold |
| ASE | C3/C4 | Photosynthesis type |
| LDAYCRIT | | Threshold for senescence (hours of daylight) |
| **SL_EMECROP** | - | Additional SLA at emergence. |
| **FF_R** | - | Fraction of biomass allocated to fruit |
| **SO_CROP** | - | Fraction of biomass allocated to stem |
| **MAX_HEIGHT** | m | Maximum crop height |

**Table 1**. Illustrating some of the most important crop-specific parameters necessary to run T&C-CROP.

## 2.4 Description of selected sites and validation data

It is crucial to model agricultural fields which experience both monocropping and crop rotations, as these practices are significant and widespread (Eurostat, 2020). This modelling approach
also serves as an excellent benchmark for complex mechanistic crop models such as T&C-CROP. An important objective was to select sites with on-site observational records that could demonstrate T&C-CROP's capability to continuously simulate field growth across various rotation and management practices within a single simulation. This contrasts with the common practice of starting a new simulation for each crop individually. The benefit of a continuous model
simulation is that this allows T&C-CROP to account for legacy soil conditions, including soil moisture, soil carbon, based on historical management practices—such as crop residue management, fertilizer application, and irrigation. This approach ensures our model accurately reflects the cumulative impact of past agricultural practices on current and future crop performance.



To showcase T&C-CROP's capabilities, we selected four well-monitored agricultural sites, all characterized by a temperate climate but featuring diverse cropping systems and management
practices. These sites are affiliated with FLUXNET (Heinesch et al. 2021) and have been previously utilized for model evaluations (e.g., Boas et al. 2021), making them ideal for model intercomparison and benchmarking. Further details about the selected sites are provided in Table 2.

| Site | Crops | Years Simulated | Further site specific info | FLUXNET Link |
|---|---|---|---|---|
| CH-OE2 (Solothurn, Switzerland) | Wheat, Barley, Grass, Potato, Rapeseed, Peas. (Rainfed) | 2004-2020 | Dietiker et al. (2010); Ecosystem Thematic Center (2021). | Site Info for CH-Oe2-AmeriFlux (fluxnet.org) |
| CH-CHA (Zug, Switzerland) | Grass (Rainfed) | 2006-2015 | Hörtnagl et al. (2018) | Site Info for CH-Cha-AmeriFlux (fluxnet.org) |
| US-NE1 (Nebraska, USA) | Maize (Irrigated) | 2002-2013 | Suykeret al. (2004) | Site Info for US-Ne1-AmeriFlux (fluxnet.org) |
| BE-LON (Valonia, Belgium) | Sugar Beet, Wheat, Potatoes, Mustard (cover crop), Maize, Oat. (Rainfed) | 2004-2020 | Dufranne et al. (2011), Buysse et al 2017, Moureux et al. 2006; Dumont et al. 2023 | DOI Listing for BE-Lon - FLUXNET |

**Table 2**. Information regarding the agricultural sites used in this study.

## 2.5 Model Intercomparison

The performance of T&C-CROP was compared with that of three other leading similar models which have been previously validated on the same sites. Specifically, JULES-CROP was evaluated on the US-NE1 site for maize, CLM-CROP on the BE-LON site for sugar beet, potatoes, and wheat, and ORCHIDEE-CROP on the BE-LON site for wheat. The data for this
comparison was extracted from published works: Williams et al. (2017) for JULES-CROP, Boas et al. (2021) for CLM-CROP, and Wu et al. (2016) for ORCHIDEE-CROP. An open-source web-





based tool, available at https://github.com/ankitrohatgi/WebPlotDigitizer, was used to extract numerical data from plot images provided in the publications. Minor discrepancies due to the accuracy of the graph digitizer are expected.

# 3. Results

## 3.1 Land surface Energy balance

Across the four selected sites, the model captured the monthly trends in energy fluxes as illustrated in Figure 1. The mean monthly $r^2$ across sites for net radiation (Rn), sensible (H) and latent heat (QE) was 0.97, 0.85 and 0.96 respectively (Full table available in Supplementary 4). Unpacking this further the across Rn, H and QE mean daily $r^2$ was 0.68 which is commendable given potential discrepancies in the energy budget closure of flux tower measurements.

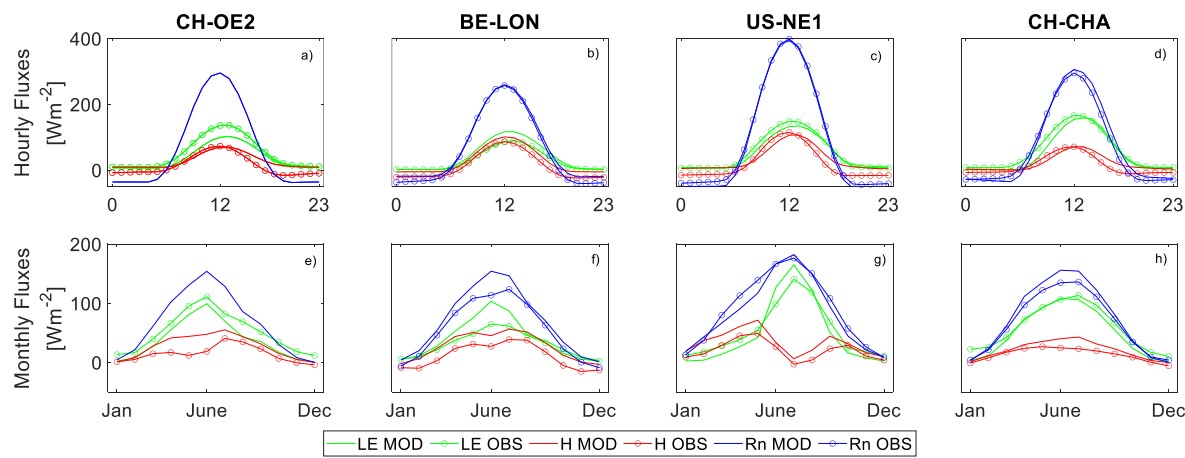

**Figure 1**. This graph illustrates the comparison between modelled and observed energy fluxes across various sites: CH-CHA (grassland), US-NE1 (maize), CH-OE2, and BE-LON (both with complex crop rotations). The hourly fluxes, representing the average diurnal cycle, are depicted with different colours: green for latent heat flux (LE), red for sensible heat flux (H), and blue for net radiation (Rn).



## 3.2 Gross Primary Productivity, Ecosystem Respiration, Net Ecosystem Exchange and Soil Moisture.

We found that to capture the correct timing of GPP fluxes for each crop (Figure 2) it was
395 imperative to draw on a traits based approach, as lumping different crops into PTFs performed
significantly worse. As illustrated in Figure 2, the magnitude and timings of the GPP fluxes are
correctly captured, as are the differences between crops and to a lesser extent between
seasons (same crop different year). Additionally, in Table 4 the modelled and observed seasonal
sum of gross primary productivity (GPP), ecosystem respiration (RECO) and their difference;
net ecosystem exchange (NEE) is presented; a season is defined as the period between crop
emergence to harvest. T&C-CROP was able to capture the seasonality of GPP, across crops,
within roughly a 10% range of observed values, as depicted in Table 3. Although it did slightly
less well at capturing seasonal RECO (Table 3).

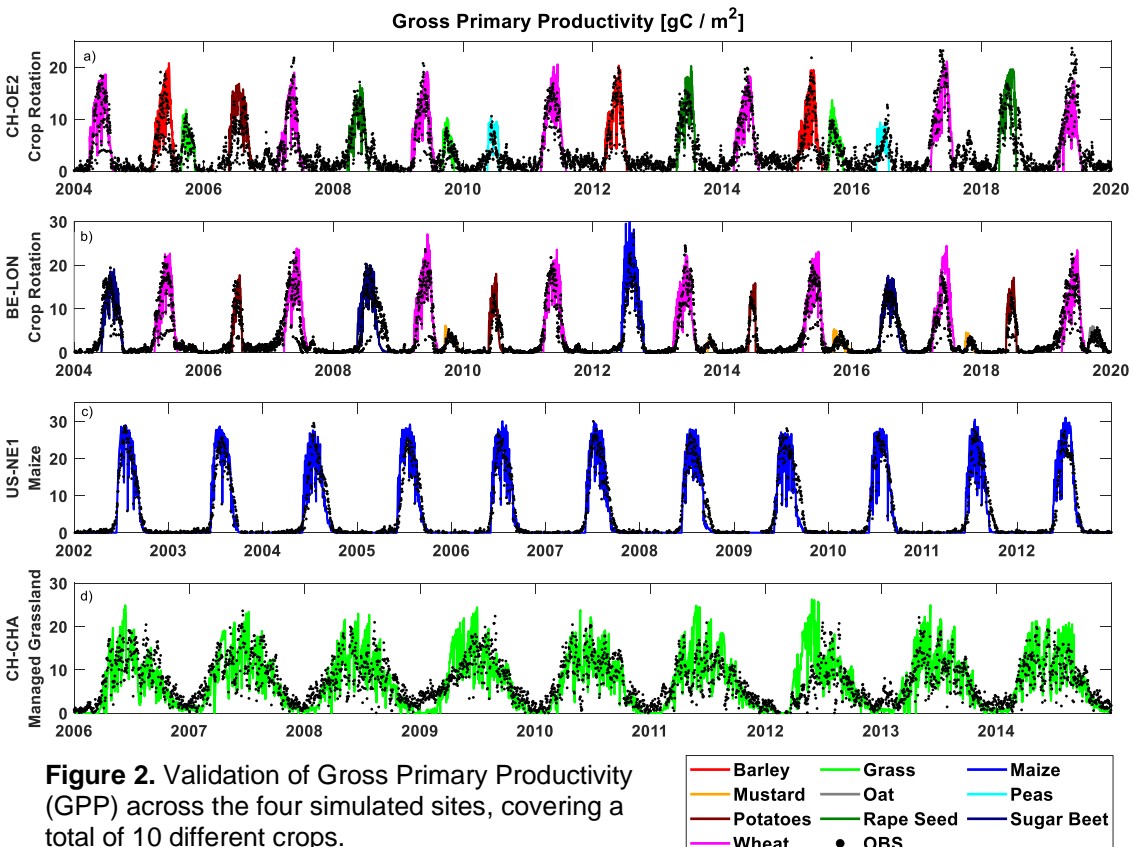

**Figure 2.** Validation of Gross Primary Productivity (GPP) across the four simulated sites, covering a total of 10 different crops.





| CH-OE2: Crop Averages | | | | | | | | |
|---|---|---|---|---|---|---|---|---|
| CROP | MODGPP | OBSGPP | Δ (%) | MODRECO | OBSRECO | Δ(%) | MODNEE | OBSNEE |
| Wheat | 1153 | 1300 | -11 | 722 | 751 | -4 | -431 | -504 |
| Barley | 1127 | 1069 | 5 | 662 | 575 | 15 | -465 | -408 |
| Cover | 433 | 414 | 5 | 294 | 308 | -5 | -139 | -75 |
| Rape Seed | 1254 | 1098 | 14 | 749 | 888 | -16 | -505 | -366 |
| Peas | 377 | 386 | -2 | 187 | 527 | -65 | -190 | -366 |
| Potato* | 1477 | 935 | 58 | 772 | 980 | -21 | -706 | 199 |
| AVG | | | 9 | | | 10 | | |

**Table 3a** * Note that potatoes were crop failure event due to hail. A phenomenon we currently do not simulate; this crop was discarded from the averages. Also note that AVG is absolute average. Values are seasonal, from date of sowing to harvest.

| BE-LON: Crop Averages | | | | | | | | |
|---|---|---|---|---|---|---|---|---|
| Crop | MODGPP | OBSGPP | Δ (%) | MODRECO | OBSRECO | Δ(%) | MODNEE | OBSNEE |
| Sugar Beet | 1353 | 1455 | -7 | 537 | 664 | -19 | -816 | -808 |
| Wheat | 1526 | 1496 | 2 | 801 | 887 | -10 | -725 | -570 |
| Potato* | 531 | 556 | -5 | 236 | 454 | -48 | -294 | -149 |
| Mustard | 192 | 162 | 19 | 94 | 204 | -54 | -99 | 43 |
| Maize | 1876.3 | 1492.9 | 25.7 | 951.8 | 963.2 | -1.2 | -924 | -595.4 |
| Oat | 280 | 288 | -2 | 169 | 299 | -43 | -168 | 16 |
| AVG | | | 11 | | | 31 | | |

**Table 3b** *Note that a defoliant was applied to potatoes mid-season, a management which was incorporated into T&C-CROP.

| US-NE1 | | | | | | | | |
|---|---|---|---|---|---|---|---|---|
| Crop | MODGPP | OBSGPP | Δ (%) | MODRECO | OBSRECO | Δ(%) | MODNEE | OBSNEE |
| Maize | 1785 | 1668 | 7 | 731 | 1161 | -37 | -1054 | -566 |

**Table 3c** Values are the average of all periods from sowing to harvest (2002-2012)

| CH-CHA | | | | | | | | |
|---|---|---|---|---|---|---|---|---|
| Crop | MODGPP | OBSGPP | Δ (%) | MODRECO | OBSRECO | Δ(%) | MODNEE | OBSNEE |
| Grass | 708 | 763 | 12.7 | 612 | 560 | 57 | -156 | -58 |

**Table 3d** These values are the average all the periods running from sowing to harvest for which we had site data (2006-2020).



T&C-Crop's skill in simulating Soil Water Content (SWC) is illustrated in Figure 3. The maize monoculture site (US-NE1) along with the crop rotation site (BE-LON) were chosen for this illustration due to their long observational SWC record. At a depth of 25cm, a correlation coefficient of $r^2$ =0.64 was achieved between daily observed and modelled SWC at the US-NE1 site, a similar value of 0.62 is achieved at the BE-LON site (if we only include data until the sensor change in 2015).

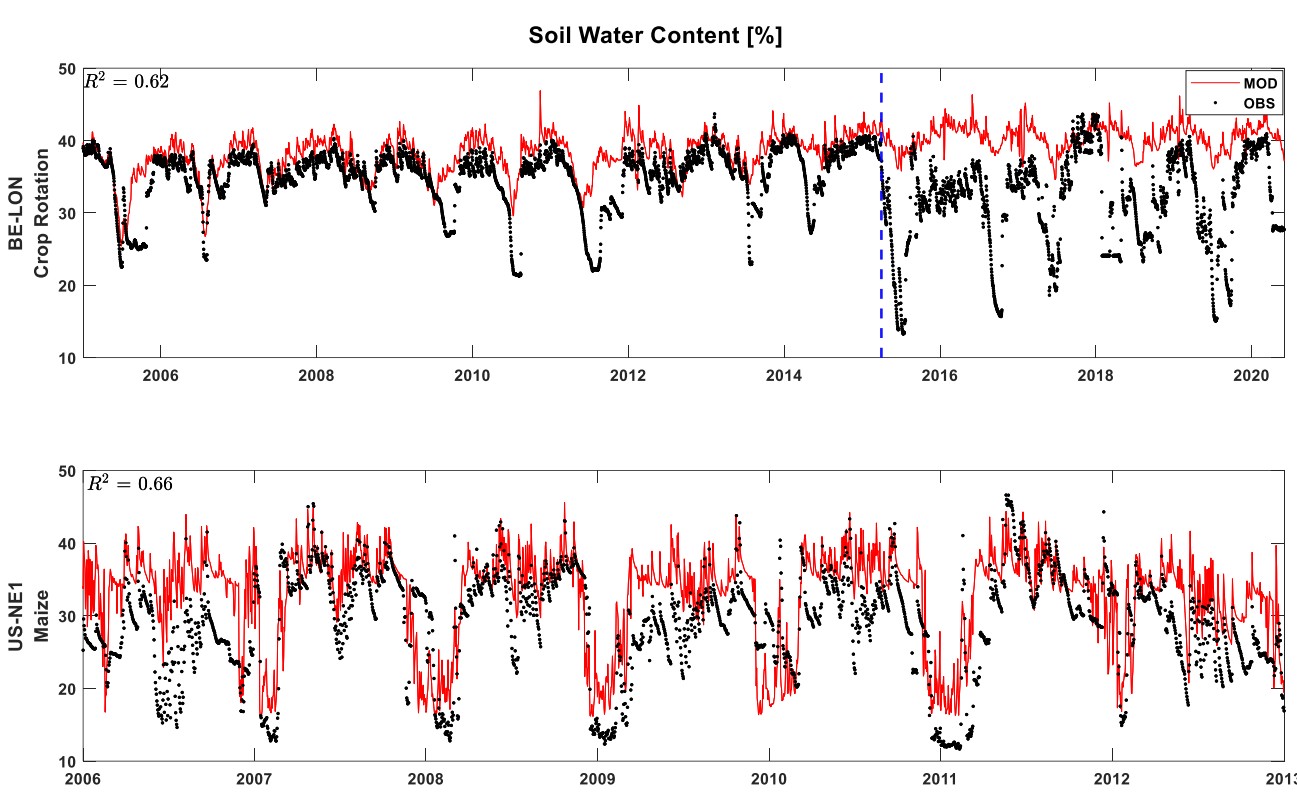

**Figure. 3** Validation of Soil Water Content (SWC) across BE-LON, complex crop rotation and US-NE1, maize monoculture. Both sites represent modelled and observed SWC at a depth of 25cm. The dashed blue line represents the date of a sensor change.





## 3.3 Crop development: LAI and Biomass Growth

T&C-CROP was able to capture the timing of leaf flushing and growing season length across various simulated sites and crop types (Figure 4). The model demonstrated considerable skill

in reproducing peak season Leaf Area Index (LAI), indicated by a correlation coefficient ($r^2$) of 0.75, 0.66 and 0.61 for CHOE2, BELON and USNE1 respectively. However, on CH-CHA, grassland site, whilst the leaf growth pattern was clearly captured, there was no significant correlation between observed and simulated peak LAI, likely due to the spread in recorded LAI values on each date. Importantly, T&C-CROP successfully captured most differences in LAI

among different crops; most clearly depicted with mustard and wheat at the BE-LON site (Figure 4, panel b). The model's strongest performance was in replicating LAI dynamics at the US-NE1 maize monoculture, achieving an $r^2$ of 0.77, a satisfactory result considering the limited developments to T&C-CROP and inherent heterogeneity in field-based LAI sampling and different cultivars sown.

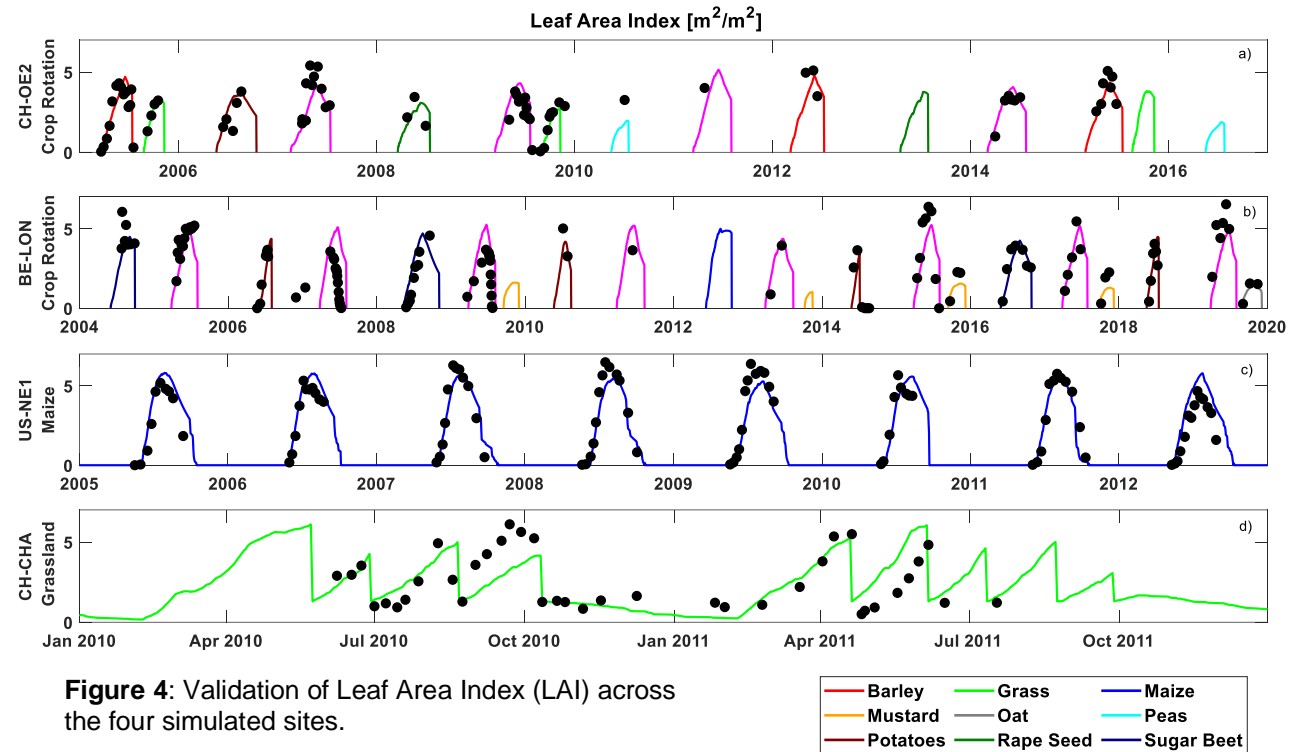

**Figure 4**: Validation of Leaf Area Index (LAI) across the four simulated sites.



The validation of T&C-CROP against observed crop harvests (Table 4) demonstrates the model's ability to accurately capture biomass differences at harvest time among various crops and effectively partition assimilated carbon into different crop components, such as stems and grains. Across the four simulated sites, T&C-CROP successfully predicted the annual harvested aboveground biomass (AGB) within approximately 20% of the observed values, with a few exceptions (Table 4).

We also assessed dynamic carbon allocation mechanisms throughout the growing season at the US-NE1 site, using published observations (Peng et al. 2018) as a reference (Figure 5). Our findings indicate that T&C-CROP effectively captures the overall trend and magnitude of carbon allocation to specific crop components such as leaves, stems, and grains. This underscores the model's promising ability to represent the dynamic processes that drive crop growth and development. Regarding Figure 5, it is important to note that in 2007 at the US-NE1 site, our modelled above-ground carbon (AGC) was slightly lower than observed, peaking at 9.5 t C/ha compared to the observed 11.34 t C/ha (Fig. 7a).

We analysed crop rotations at two sites, CH-OE2 and BE-LON, and also evaluated T&C-CROP's performance on maize at the US-NE1 site and grassland at the CH-CHA site. At the CH-OE2 site, we simulated 19 crop cycles over fifteen years (2004-2019). On average, the harvested aboveground biomass (AGB) was simulated within 10% of recorded values. Grain and straw were simulated within 13% and 30% of recorded values, respectively. However, inter-annual variation in crop growth and carbon allocation to different pools (grain/straw) were difficult to capture.

At the BE-LON site, we simulated 21 crop cycles over sixteen years (2004-2020). Winter wheat and maize were well simulated, with AGB and grain values, on average, within 10% of observations. Straw was slightly overestimated, by 27% for wheat and 13% for maize. If we account for crop residues, particularly the first few centimetres of straw, our simulated values





could align more closely with observed values. Additionally, including the belowground component of sapwood, which is currently excluded, would likely bring simulated AGB values even closer to observations. For wheat, the average residue at BE-LON was 26% of AGB, with a standard deviation of 4%. Potatoes at BE-LON were more challenging to simulate accurately,

partly due to the defoliant treatment applied in mid-August, which is not currently included in our model. This resulted in simulated tuber biomass (daughter tubers) being about 50% lower than observed.

Over eleven years (2002-2012) at the US-NE1 site, simulated maize yield (kernel) was within

8% of recorded values on average. For the grassland site CH-CHA, harvest data was available for eight cuts from 2008 to 2010. Here simulated harvested biomass was within 20% of recorded values on average. Full results in a tabular format are included in supplementary 4.

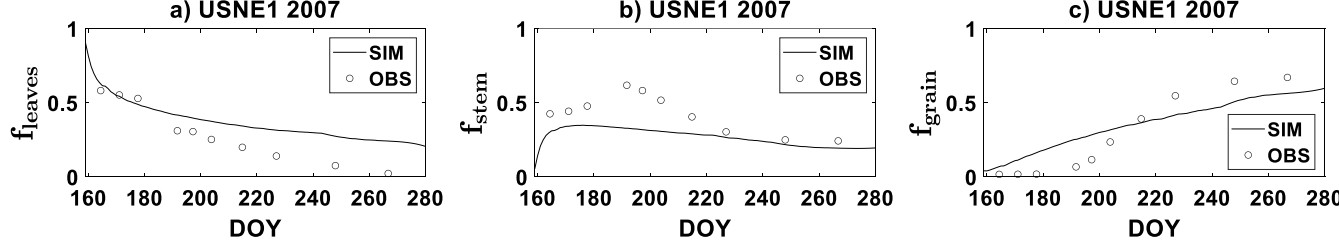

**Figure 5** Total fraction of above-ground biomass in leaves, stems, and grain at the maize site (USNE1), illustrating the partitioning of assimilated carbon by T&C. Leaves are represented by the "foliage" pool, stems include sapwood and dead sapwood pools, and grain consists of carbohydrate reserves, fruit and flower pools. Observed values are derived from the graphs in the supplementary material of Peng et al. (2018).



| CH-OE2 Yields | | | | | | | | |
|---|---|---|---|---|---|---|---|---|
| Crop | OBS AGB | SIM AGB | Δ (%) | OBS STRAW | SIM STRAW | Δ (%) | OBS GRAIN | SIM GRAIN | Δ (%) |
| Wheat | 4.3 | 3.7 | 14.0 | 1.7 | 1.3 | 23.5 | 2.6 | 2.4 | -7.7 |
| Barley | 3.9 | 3.9 | 0.0 | 0.7 | 1.2 | -71.4 | 3.2 | 2.7 | -15.6 |
| Rape Seed | / | / | / | / | / | / | 2.0 | 2.2 | 10 |
| Peas | / | / | / | / | / | / | 3.5 | 6.1 | 74.3 |

| BE-LON Yields | | | | | | | | |
|---|---|---|---|---|---|---|---|---|
| Crop | C Exported | SIM AGB | Δ (%) | OBS STRAW | SIM STRAW | Δ (%) | OBS GRAIN | SIM GRAIN | Δ (%) |
| Sugar Beet | / | / | / | / | / | / | 8.9 | 6.9 | -22.5 |
| Wheat | 5.5 | 5.9 | -6.0 | 1.8 | 2.5 | -27.0 | 3.7 | 3.5 | -5.4 |
| Potato | / | / | | / | / | / | 3.3 | 2.2 | -33.3 |
| Maize | 7.8 | 7.2 | 7.1 | 3.6 | 4.2 | -13.4 | 4.2 | 4.2 | 0.0 |

| US-NE1 Yields | | | | | | | | |
|---|---|---|---|---|---|---|---|---|
| Maize | / | / | / | / | / | / | 5.5 | 4.9 | -10.9 |

| CH-CHA | | | | | | | | |
|---|---|---|---|---|---|---|---|---|
| Grass | 0.85 | 1.00 | 17.6 | / | / | / | / | / | / |

**Table. 4**. In T&C-Crop, crop carbon is distributed across six distinct biomass carbon pools: B1=Foliage, B2=Living Sapwood, B3=Fine Roots, B4=Carbohydrate Reserves, B5=Fruit and Flowers, and B6=Standing Dead Foliage. In Table 4, Simulated Above Ground Biomass (AGB) corresponds to the sum of all T&C-Crop's biomass pools excluding B3 (Fine Roots); we assume that all sapwood is aboveground, an approximation which is reasonable for most crops. Simulated Grain is represented by the sum of B5 (Fruit and Flowers) and B4 (Carbohydrate Reserves), which are expected to be contained mostly within the fruits for a crop, and simulated straw is derived from the sum of B1 (Foliage), B2 (Living Sapwood), and B6 (Standing Dead Foliage). Validation for belowground biomass (roots) was not possible due to the absence of on-site data. Note that for US-NE1, a value of 43%, as suggested by the PI, was used to translate t ha $^{-1}$ to t C ha $^{-1}$. For CH-CHA grass yields are annual from 2008-2010. * Note that in CH-OE2 OBS AGB refers to the total AGB at the time of harvest whereas in BE-LON C Exported refers to the harvested component of the AGB. All values are in t C ha $^{-1}$.





## 3.5 Model Intercomparison

T&C-CROP simulations were compared to those of JULES-CROP (Williams et al. 2017). Figures 6 and 7 illustrate how both models, relative to each other represent AGB and LAI over a course of eight years for a Maize (US-NE1) field. Despite T&C-CROP being arguably more

process-based and more parameter parsimonious, both models did a comparable job at capturing the correct magnitude and timing of LAI and AGB, neither model correctly simulated inter-annual variations in peak LAI or AGB.

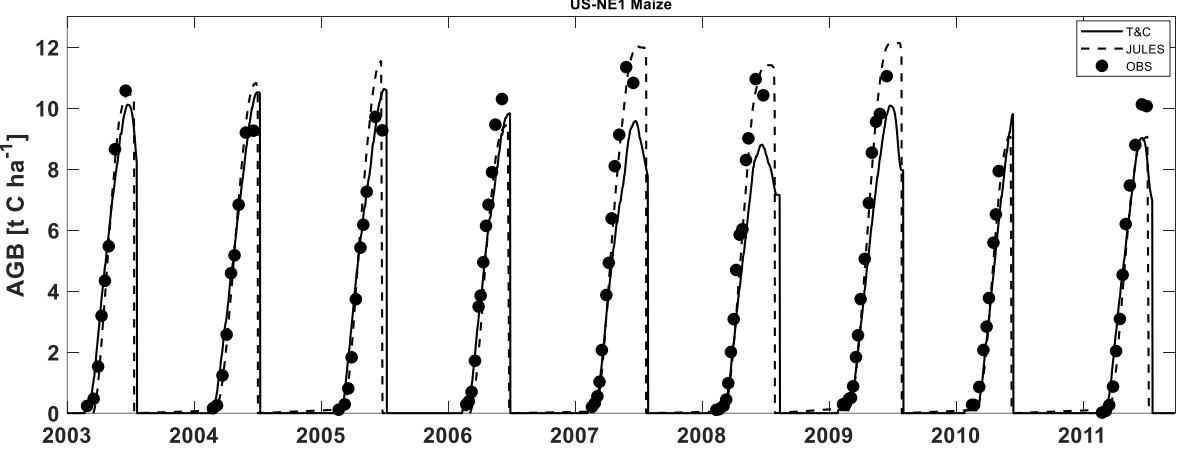

**Figure 6** Simulation of above ground biomass by both T&C-CROP and JULES-CROP models on the US-NE1
Maize site.

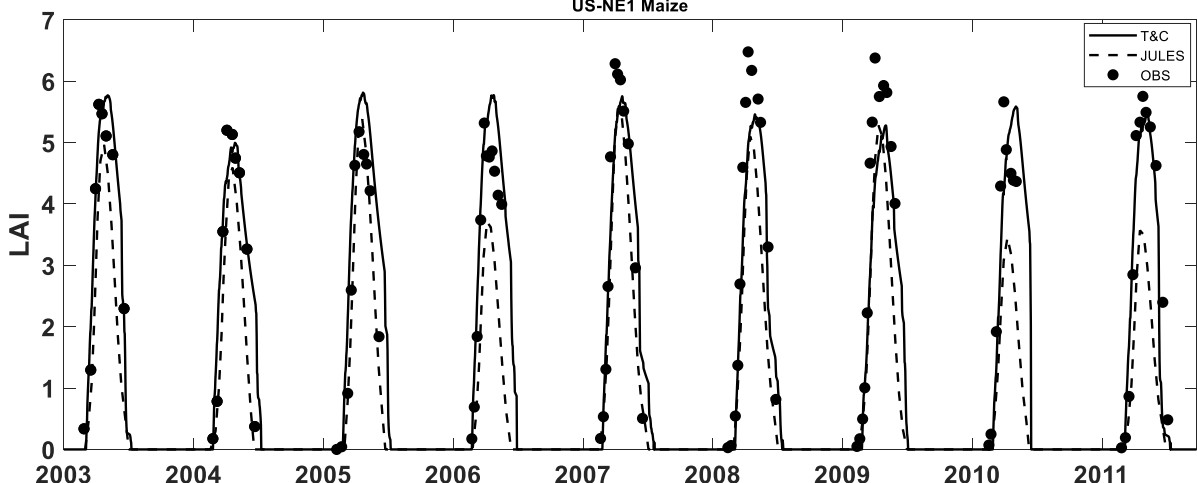

**Figure 7** Simulation of LAI by both T&C-CROP and JULES-CROP models on the US-NE1 Maize site.





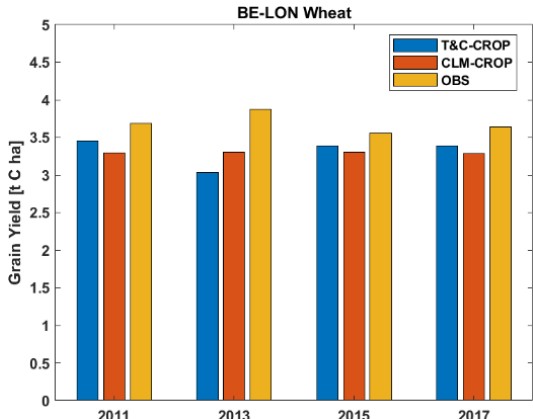

**Figure 8.** Side by side comparison of CLM-CROP and T&C-CROP.

T&C-CROP simulations conducted over the crop rotation site BE-LON were compared to those of CLM-CROP (Boas et al. (2021). Figure 8 illustrates how both models simulate grain yields for winter wheat across the four years which were presented in the CLM-CROP paper. To produce this comparison, we converted CLM-CROPS' modelled values, which are reported in T DM ha$^{-1}$ to T C ha$^{-1}$ using the average site-reported C content per unit of dry mass for wheat grain during these four years which was 40.5%; there was little interannual variation in this value, (<3%). Unfortunately, there is not sufficient data or variation in grain yield to truly assess the efficacy of either model, however, based on the presented observations, both capture the correct magnitude but neither capture the inter-annual observations in yield. Figure 9 and 10 illustrate how both models successfully represent LAI as well as key land surface fluxes over the years for which sugar beet and potatoes were sown. Note that a defoliant was applied to potatoes at the BE-LON site (Aubinet et al., 2009). To replicate this in T&C-CROP, we simulated a sudden "cut" on the recorded date of defoliant application.





**BE-LON Sugar Beet (2008 and 2016)**

**Figure 9** Simulation of Leaf Area Index (LAI), Net Ecosystem Exchange (NEE), latent heat flux (LE),sensible heat flux (HE) and net radiation (Rn) across both T&C-CROP and CLM-CROP for Sugar Beet cultivated at the BE-LON site in 2008 and 2016.

**BE-LON Potatoes (2010,2014,2018)**

**Figure 10** Same as figure 9 but for Potatoes.






Lastly, T&C-CROP was evaluated against results from ORCHIDEE-CROP (Wu et al. 2016) for the winter wheat season on the BE-LON site in 2006 (Figure 11).

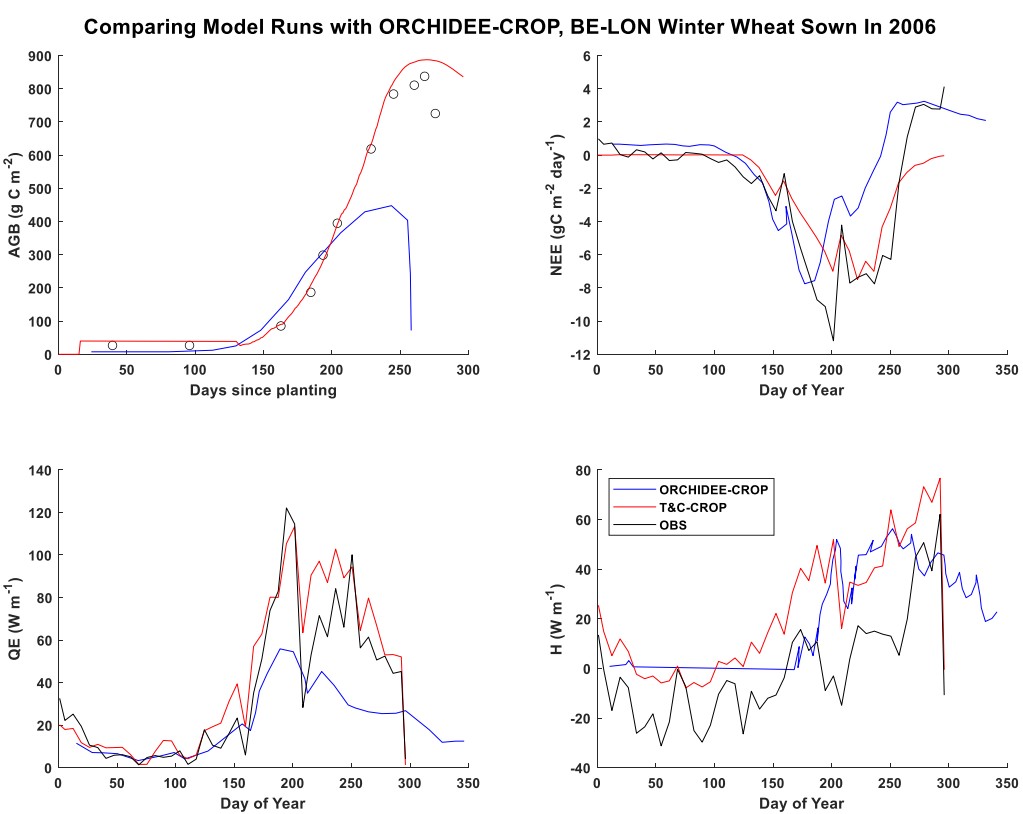

**Figure 11.** Illustrating a comparison of ORCHIDEE-CROP outputs from Wu et al. 2016 and T&C-CROP outputs from this paper for Winter Wheat sown in BE-LON. Note that both Latent (QE) and Sensible Heat (H) were smoothed using a weekly time step to improve graph readability. Note AGB here refers to total, not only harvestable AGB.

ORCHIDEE-CROP (Wu et al. 2016) undershoots above ground biomass (AGB) by about 50% whilst T&C-CROP does a much better job, albeit overshooting AGB by just under 10% (Figure 10). More specifically, T&C-CROP achieved a correlation coefficient of $r^2 = 0.94$ between simulated and observed AGB whilst this was 0.2 for ORCHIDEE-CROP.



## 4. Discussion

The integration of three new crop-specific parameters, combined with streamlined model developments, has significantly enhanced the representation of cropland sites in T&C-CROP. Our findings include the successful validation of over ten different crops sown in four
heterogeneous agricultural fields, varying in both management practices and climate conditions. Results also demonstrate that T&C-CROP performs comparably to other leading terrestrial biosphere models (TBMs) without having to increase model complexity or introduce crop-specific carbon pools. This underscores the effectiveness of T&C-CROP as a highly parameter-efficient and process-based model for future studies.


This improved incorporation of croplands into T&C opens new avenues for modelling land-surface interactions, hydrology, carbon fluxes, and crop yields. For instance, the enhanced representation of sensible heat (H), latent heat (LE), and net radiation (Rn) facilitates research on land surface interactions. Similarly, better modelling of evapotranspiration (ET) and leaf area
index (LAI) benefits hydrological studies, while improved accuracy in net ecosystem exchange (NEE) and soil carbon storage could aid contemporary carbon emission mitigation efforts. Hydrological and carbon storage implications of land-use transitions involving crop, forest, pasture conversion, as well studies on optimal irrigation and fertilization application in a changing climate are among the foreseen applications of T&C-Crops.


Additionally, beyond the biomass, hydrological and energy balance metrics validated in the results section, T&C-CROP can also simulate belowground soil biogeochemical dynamics (Fatichi et al., 2019). We have included some outputs for illustrative purposes (Supplementary 6). T&C-CROP captures changes in nutrient leakage as a function of local weather, crop type,
fertilizer regime, and legacies. Using the biogeochemistry module, we identified a boost in microbial carbon post-harvest, nutrient flushing following fertilization, and predominantly after rainfall events.





The utility of a versatile tool like T&C-CROP is intended to perform at a regional spatial scale.
However, validating its efficacy at this spatial level poses significant challenges due to sparse comprehensive data and the multitude of factors influencing crop growth, including socio-economic variables. However, many of the issues we encountered during site-level validations are expected to diminish at a broader scale, as local variations average out and climatic variables assume greater importance. For instance, representing microscale field management 610 proved challenging during validation efforts. Adjusting for different cultivar types, accurately determining crop-specific carbon allocation parameters, implementing practices such as the use of growth regulators or defoliant/fungicide treatments at sites like BE-LON (Dugranne et al. 2011) or dealing with hail at CH-OE2 (Revill et al. 2016) proved complicated. Furthermore, T&C-CROP struggled with simulating post-harvest processes, likely due to inadequate 615 knowledge of post-harvest management practices such as residue management and soil preparation/tillage. All the above considerations are primordial at the field scale, but are likely to exert less influence on crop growth across larger spatial scales, where climatic conditions are expected to play a dominant role.

## 5. Conclusion


T&C-CROP was introduced to enhance T&C's representation of croplands and associated carbon, energy and nutrient fluxes. In this study we have assessed the extent to which T&C-CROP accurately depicts crop growth and associated land surface fluxes across four distinct agricultural sites CH-OE2, BE-LON, CH-CHA, US-NE1. Each site was subject to varying 625 management practices such as irrigation, fertilizer and defoliant application and had several types of crops, either as a monoculture or as a crop rotation scheme. Our model validation covers over 50 years and 61 crop cycles, encompassing more than nine staple crops and also included comparison with results from other leading TBMs.





This study demonstrates how with minimal model structural changes and only three additional parameters, it is possible to accurately represent Gross Primary Productivity (GPP), LAI (Leaf Area Index) and organ-specific harvests not only in monocultures but also in sites with complex crop rotations and diverse management practices. Of particular novelty we adapted the carbon allocation scheme for crops and implemented a novel routine which allowed for multiple

cropping cycles within one calendar year within the same model run. This enhancement enables more realistic simulations of field dynamics.

Our approach with T&C-CROP is grounded in practical utility. While our validation efforts were thorough, they were not overly fixated on meticulously simulating variables such as yield,

considering that this is only one of the many model outputs. We were realistic with limitations in parameter constraints as a high-level granularity was not a primary objective. We prioritized broad applicability over micromanagement details like cultivar choice, which is unlikely available at larger scales.

T&C-CROP's research horizon is to explore in a single model the effects of various crops on yields, energy dynamics, and carbon fluxes, as well as assessing how major climatic factors (temperature, precipitation, $CO_2$, relative humidity, etc.) interact with management practices (fertilizer, irrigation) to influence crop yields but also byproducts such as nutrient runoff, soil degradation, and carbon sequestration. These latter points being particularly valuable for

research aimed at assessing climate risk in agriculture.

Future studies with T&C-CROP are envisioned to be conducted over broader spatial scales, where detailed management practices or specific cultivar information are less important. T&C-CROP's ability to capture geographical differences induced by climate and soil properties are

expected to overshadow local variations due to specific cultivars or management practices. This capability makes it an invaluable tool for understanding and predicting large-scale environmental patterns and their implications.



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

**Author Contribution:**

JBP and AP designed the project and carried out the simulations. SF and AP are the main developers behind T&C, with modifications for T&C-CROP made by JBP, SF and AP. JBP prepared the manuscript with contributions from all co-authors.

**Competing Interests:**

The authors declare that they have no conflict of interest.

**Acknowledgements:**

Thank you to the respective PI's and data managements across the four sites who made data available.