# Peer review of "T&C-CROP: Representing mechanistic crop growth with a terrestrial biosphere model (T&C, v1.5): Model formulation and validation."

_EGUsphere, 2024_

## Author Comment (AC1)

**Dear Editor,**

We would like to thank you and the reviewers for their overall positive evaluations of the manuscript. Both reviewers raised very valid and important comments and provided excellent suggestions for improving our manuscript. In the revised version, we have taken them fully under consideration and improved the manuscript accordingly.

Following this, please find attached a detailed point-to-point response to all comments raised by the reviewers. The line numbers from the document with track changes are used to direct the reader towards the specific comments addressed.

**Best wishes,**
Jordi Buckley, Simone Fatichi, Bonnie Waring, Athanasios Paschalis
* * *
**Response to Reviewer 1**

**Comment:**
*This is a well-written and interesting paper that expands the T&C model to simulate crop growth based on physiological crop attributes. The T&C model is a well-established and validated biosphere model and T&C-CROP enhances its capabilities. Its performance is comparable to other similar models, but T&C-CROP has notable advantages of requiring few parameters, being able to simulate multiple crop cycles (contrary to other models that need to be re-initialized for each cycle), and carrying soil legacy information, which influences crop growth, thanks to its integration within T&C model. I recommend publication after the following comments are considered.*

**Response:**
We would like to thank the reviewer for their overall positive evaluation of our work.
* * *
**Comment:**
*It would be useful to include a simple schematic of the T&C-CROP model that particularly emphasizes the new contributions introduced here. T&C is well established, so there is no need for a full model diagram. I leave it to the authors to find the right balance in providing the details needed to highlight the new aspects.*

**Response:**
We agree with the reviewer's comment and have added a simple schematic illustrating the transition from T&C to T&C-CROP. This has been included in the methodology section as Figure 1. (Line 250). Additionally, we have made it clearer in Table 1 (Line 345) by indicating that the new crop parameters are listed in bold.
* * *
**Comment:**

*During model evaluation, the parameters were adjusted within a ±30% range, following a manual trial-and-error calibration. While I understand that a systematic calibration is beyond the scope of this work, I wonder if the authors foresee incorporating calibration techniques, particularly Bayesian techniques (Markov Chain Monte Carlo calibration, for example), into the modeling framework.*

**Response:**

Thank you for this insightful comment. This is indeed a very interesting avenue for future research, although it is currently computationally challenging. Implementing Bayesian techniques in this context would require substantial computational resources, particularly given the complexity and high dimensionality of a crop model such as T&C-CROP. As a first step, a formal sensitivity analysis of T&C-CROP would be needed to identify the parameters most sensitive to crop growth, which could then guide further efforts. While this is beyond the scope of this introductory model description, we appreciate the suggestion and have now mentioned this outlook in the conclusion (Line 685-690).
* * *
***Comment:***

*In the introduction, I appreciated the discussion regarding the importance of further developing physics-based models, despite having sometimes lower performance than machine-learning approaches. As more data becomes available, probably an integration of data-driven and process-based approaches could help significantly improve our prediction capability.*

**Response:**

Thank you for this positive feedback; we are glad to see that this viewpoint is echoed. This comment regarding the future potential for data-model fusion is highly valid; accordingly, we have incorporated this point into our final outlook section, as reflected in Lines 664-666.
* * *
**Comment:**

*In the model intercomparison, providing additional details on how the other models were set up for simulations would be helpful. For example, were they calibrated to improve performance? If so, what method was used? Including more details would allow for a fairer comparison I believe.*

**Response:** Thank you for the feedback. We have now included an additional paragraph (Line 380-L393) in Section 2.5, *Model Intercomparison,* to provide further information as suggested.

**Comment:**

*In the discussion, I am curious whether the authors plan to further develop T&C-CROP. In an agricultural context, other processes such as soil erosion, pesticide applications, as well as other agricultural practices, especially those that are becoming more common with climate-smart agriculture. While this model represents a step forward from other biosphere models mentioned in the manuscript (ORCHIDEE, JULES, etc.), other models specifically developed for agricultural ecosystems (APEX, DNDC, etc.) have made significant advancements in parameterizing ag practices, even though they may be less physically-based in approaching other biosphere processes.*

**Response:** This is a very interesting point. We have recently been examining data from long-term field experiments (e.g., Rothamsted) to validate the incorporation of additional agricultural practices in T&C-CROP. However, before fully pursuing this, our envisaged future work will focus on improving the representation of nutrient runoff, soil degradation, and carbon sequestration within agricultural fields, as well as including common practices such as tillage. We have amended our manuscript, Lines 645-654, further acknowledging the importance of the above reviewer comments.

**Response to Reviewer 2**

**Comment:**

*I would like to begin by commending the authors for their excellent work on this paper. It is very well-written, with clear language and a logical structure that makes the content easy to follow. The study introduces the crop representation into the T&C model. The methodology is presented in a highly organized manner, and the evaluation is thorough, providing convincing evidence for the effectiveness of the proposed approach. One aspect I particularly appreciated was the detailed explanation of the experimental setup and the comprehensive evaluation of the model results.*

**Response:**

We would like to thank the reviewer for their positive evaluation of our work.

**Comment:**

*However, I encountered some challenges in fully grasping the distinction between the T&C model and the T&C-CROP model. While the paper briefly touches on the enhancements made in the code and methodology, it would benefit from a more explicit description of these new contributions.*

**Response:**

Thank you for the feedback. We have now included a schematic in the methods section (Figure 1, Line 250) that visually and succinctly illustrates the model developments involved in the transition from T&C to T&C-CROP. This schematic is intended to clarify the changes made to the model before they are discussed in further detail, and we hope it makes the developments much clearer.
* * *
**Comment:**

"What does T&C stand for?"

**Response:**

This is a great question. T&C stands for *Tethys-Chloris* as introduced in Fatichi (2012) which is also referenced in our manuscript. However, the model is now simply referred to as T&C.
* * *
**Comment:**

"You provide extensive detail in the supplement regarding fertilizer application, planting, and harvest dates. Could you extend this level of detail to include the crop-specific parameters as well? Important variables are evaluated, but I would further appreciate a comparison to net ecosystem exchange (NEE) data, as this is one of the most important variables for the carbon balance in terrestrial biosphere models (TBMs)."

**Response:**

Thank you for this comment. We have now included details regarding crop-specific parameters used in Supplementary 2. The most important crop-specific parameters are included here, although if the reader is interested in exploring any others then these are available as port of the MOD_PARAM file in our code (uploaded as part of this paper).

We agree, a comparison to NEE data is very useful – this is included in Table 3 (Lines 455-460) alongside seasonal values (OBS vs MOD) for GPP and RECO. Similar to results found by similar studies we struggled to replicate RECO, particularly just after harvest, this can partly be explained due to a lack of knowledge of post-harvest field management – ploughing, crop residue etc... it should also be noted that there is often considerable uncertainty in observed fluxes and these are not always perfect by any stretch. A comment regarding this has been added to the manuscript, Line 425-426.

**Comment:**

"Line 175: Could you clarify what is meant by 'vegetation-specific approach where the model user defines the vegetation/crop in question'? This statement could benefit from further explanation."

**Response:** Thank you for the comment. We have now revised the text in Lines 179-182. We hope this makes the explanation clearer.

**Comment:**

Line 180: What is the difference between the plot-scale version and the spatially explicit version? Does the spatially explicit version involve dividing the grid into smaller sub-parts, or is there another distinction?

**Response:** This is a very good question thank you. T&C can be run in two ways: as a plot-scale version (as is done in this study and sometimes referred to as a point scale version), where topography and lateral fluxes are not explicitly considered, or in a spatially explicit manner, where a large computational domain is split into a regular grid of a common resolution of less than a hectare. Computational cells communicate laterally via lateral water flow on the surface and in the subsurface. The latter approach is computationally much more expensive and accounts for complex topography by considering local and remote solar radiation shading effects, as well as the lateral transfer of water in the surface and subsurface. The component regarding a large meshed grid has been added into the text in order to make this clearer. (Line 184)

**Comment:**

*Line 395: What is meant by "PTFs"? Please define this abbreviation for clarity.*

**Response:** We apologise for this typo, PTF is supposed to be PFT (Plant functional Type) which has now been corrected. (Line 416)

**Comment:**

*Figures 1 & 2: It would be helpful to include the $R^2$ values directly on the plots, and perhaps also the RMSE values. RMSE could even be color-coded for additional visual clarity.*

Response: We thank the reviewer for their comment. In response, we have added the R-squared ($R^2$) values to the plots in Figure 1, color-coded as suggested. Additionally, we have now included both $R^2$ and RMSE on Figure 2. Please note that, as Figure 1 has now

been converted into a schematic, the figures have been renumbered accordingly. Therefore, this comment now refers to Figure 2 and 3.
* * *
**Comment:**
Line 535: I don't fully understand the purpose of comparing T&C-Crop to JULES-CROP, especially since only LAI and AGB are compared. Comparisons with observed data would be more convincing. The same applies to the comparison with CLM-CROP on the following page.

**Response:** Thank you for this comment. A comparison between observed data (LAI and AGB) and model simulations (JULES-CROP and T&C-CROP) are provided in Figure 7 and 8. To make this clearer we have edited the caption on both of these figures.
* * *
**Comment:**
Could you clarify whether the model needs to be re-parameterized for each location? If so, how is this addressed? This point would benefit from more detail.

Response: Thank you for this comment, in Line 333 "Therefore, the model needs to be re-parameterized for certain parameters for each site." was added in order to make this clearer. This is addressed via a manual trial and error calibration (L334). We follow a traits-based approach meaning we have to input different crop parameter values for each site as each site has a different crop type and even if the same crop often a different cultivar type,  also see response to comment regarding Line 175 above (now lines 179-181).
* * *
**Comment:** *Additionally, could you elaborate on the three new crop-specific parameters? Which parameters are these?*

Response: Thank you for this comment. The three new crop-specific parameters are now clearly highlighted in bold in Table 1 and also explicitly referenced as new parameters in section 2.2. Lines, 271 and 284.
* * *
**Comment:** *I would have appreciated a global application for crop yields, even if not in direct comparison to observed data. Perhaps a comparison to other models would fit well here.*

**Response:** Thank you for this comment. Indeed, this is an excellent idea. While this paper focuses on introducing the model developments, accompanied by site-level validations and benchmarking against other models, we are currently working on

expanding T&C-CROP to larger spatial scales and exploring various options for achieving this effectively. We touch on this briefly in Lines 664-667.